# Development, Validation and Preliminary Experiments of a Measuring Technique for Eggs Aging Estimation Based on Pulse Phase Thermography

**DOI:** 10.3390/s22093496

**Published:** 2022-05-04

**Authors:** Antonino Quattrocchi, Fabrizio Freni, Roberto Montanini, Simone Turrisi, Emanuele Zappa

**Affiliations:** 1Department of Engineering, University of Messina, c.da di Dio, Vill. Sant’Agata, I-98166 Messina, Italy; antonino.quattrocchi@unime.it (A.Q.); fabfreni@hotmail.com (F.F.); roberto.montanini@unime.it (R.M.); 2Department of Mechanical Engineering, Politecnico di Milano, Via La Masa 1, I-20156 Milano, Italy; simone.turrisi@polimi.it

**Keywords:** egg freshness, pulse thermography, pulse phase thermography, active thermography, non-destructive methods, image processing, food industry

## Abstract

Assessment of the freshness of hen eggs destinated to human consumption is an extremely important goal for the modern food industry and sale chains, as eggs show a rapid natural aging which also depends on the storage conditions. Traditional techniques, such as candling and visual observation, have some practical limitations related to the subjective and qualitative nature of the analysis. The main objective of this paper is to propose a robust and automated approach, based on the use of pulsed phase thermography (PPT) and image processing, that can be used as an effective quality control tool to evaluate the freshness of eggs. As many studies show that the air chamber size is proportional to the egg freshness, the technique relies on the monitoring of the air chamber parameters to infer egg aging over time. The raw and phase infrared images are acquired and then post-processed by a dedicated algorithm which has been designed to automatically measure the size of the air chamber, in terms of normalized area and volume. The robustness of the method is firstly assessed through repeatability and reproducibility tests, which demonstrate that the uncertainty in the measure of the air chamber size never exceeds 5%. Then, an experimental campaign on a larger sample of 30 eggs, equally divided into three size categories (M, L, XL), is conducted. For each egg, the main sizes of the air chamber are measured with the proposed method and their evolution over time is investigated. Results have revealed, for all the egg categories, the existence of an analytic relationship and a high degree of correlation (R^2^ > 0.95) between the geometric data of the air chamber and the weight loss, which is a well-known marker of egg aging.

## 1. Introduction

Eggs are one of the most nutritious foods in nature and an important component in human diet, as they present a large source of proteins, vitamins, minerals and fatty acids [1]. Unfortunately, eggs show a rapid aging process that depends on various causes. Immediately after laying, their components are naturally subjected to chemical, nutritional and functional degradations [2]. Natural aging is not the only factor that influences egg quality, indeed also a bad state of preservation and transportation, especially as temperature and relative humidity are concerned, can compromise its edibility [3]. Physicochemical and structural changes are mainly due to exchange of water and CO_2_ through shell pores and osmotic migrations of water and minerals between the albumen and the yolk through the vitelline membrane. This causes liquification of the albumen, raising of the albumen pH, flattening of the yolk and growth of the air chamber [4].

Today, the most relevant issues of the food industry are largely different from those of the past. The new business strategies are now focused on the safety and healthfulness of food products, animal friendliness, environment, etc. [5]. In this context, the systematic determination of the egg freshness has become remarkably relevant to the point that consumers perceive the variability of such a parameter as a lack of quality [6]. Natural aging is responsible for the degradation of the organoleptic and nutritional properties of the egg [7]. In addition, poorly stored or too old eggs can pose a serious health risk because they can become a source of enteritis from *Salmonella* [8].

Currently, the European Commission Regulation [9] bases its standard for hen egg marketing on the air chamber height: when an egg is motionless, it must not exceed 6 mm, except for eggs marked as “extra fresh” where the maximum allowable air chamber height is decreased to 4 mm. In the latter case, the “extra fresh” appellation is maintained for 9 days from the laying date in the case of the correct storage. The air chamber forms in the egg immediately after the laying, then it rapidly grows with aging. Water and organic liquids, which are present in the egg, evaporate due to temperature, humidity and pH level. These natural causes lead to an enzymatic decomposition process that eventually generates gas [10]. For such reason, storage conditions (temperature and relative humidity) have a fundamental role in egg preservation and still represent an important issue as eggs freshness evaluation is concerned [11].

Normally, egg freshness is determined by candling and visual observation, seeking for shell cracks as well as shape and color anomalies. This test makes use of a light source (incandescent or LED lamp), which is exploited to visualize in backlight the air chamber and the yolk appearance [12]. After laying, the egg yolk looks like a faint and motionless shadow while the air chamber is barely visible at the bottom of the egg (dull pole). Over time, the albumen thins, the air chamber increases its size and the yolk is free to approach the shell, appearing as a darker shadow [13]. Modern poultry industry tends to reject handling and candling, both due to the low level of automation and the subjective and qualitative nature of the observation [14].

Hence, in recent decades, several non-destructive methods have been proposed in order to overcome the limitations of the traditional inspection method, checking for the possibility of evaluating the quality of a complete batch (control all the eggs from a lot), and not only a sampling. Visible (VIS, wavelengths range: 400–700 nm) and near infrared (NIR, wavelengths range: 750–2500 nm) absorption spectroscopy are currently the most popular options [15]. Kemps et al. [16] showed that VIS spectral data of intact eggs contain information about its freshness. A partial least squares model was built in order to predict HU (Haugh Unit, Raymond Haugh, 1937 [17]) and pH of the albumen with correlation coefficients of 0.842 and 0.867, respectively. The application of this kind of survey involves overlapping and combining different spectrum bands, producing spectral peaks of difficult interpretation. The aforementioned limits can be overwhelmed by working in the NIR field [18], as highlighted by Schmilovitch et al. [19], who successfully measured air chamber size, weight loss and pH with R^2^ > 0.90. However, the quite high correlation coefficient refers to groups of eggs rather than single eggs. Using FT-NIR (Fourier Transform Near InfraRed) spectroscopy, Giunchi et al. [20] acquired diffuse reflectance spectra in the range of 833 to 2500 nm. After each spectral acquisition, the freshness parameters (air cell height, thick albumen heights and Haugh unit) were also destructively measured. The predictive models showed an R^2^ value of up to 0.722, 0.789 and 0.676 for air cell height, thick albumen heights and Haugh unit respectively, therefore, the diffuse reflectance FT-NIR appears to be able to discriminate shell eggs during storage. All spectroscopic techniques guarantee a rapid analysis and the simultaneous evaluation of several chemical-physical properties. However, given the difficult interpretation of the results and the need of proper calibration models for standardization, spectroscopy has not adequately taken root in the field of quality control of eggs at the farm level [21]. Aboonajmi et al. [22] exploited ultrasounds to put in relation the phase velocity with some properties of the egg (HU and air chamber height), while mean amplitudes of the signal peaks were used for estimating egg aging. The study was conducted for 5 weeks on two sample groups, at 24 ± 1 °C and 5 ± 1 °C, with relative humidity of 75% and 40%, respectively. Although the results are favorable, the technique is based on the consideration, which is not always valid and generalizable, to have a uniform and similar shell for all the analyzed eggs.

Additionally, neural networks and machine learning have been employed to infer eggs freshness, in an attempt to developing an automatic method for real-time inspection on a large scale [23]. Dutta et al. [24] used an electronic nose-based system, consisting of an array of four commercial tin-oxide odor sensors, to compare intact and damaged eggs. Four supervised classifiers (multilayer perceptron), learning vector quantization, probabilistic neural network and radial basis function network, were used to classify the samples into three observed states of freshness. Automatic classification into one of the three states was achieved with up to 95% accuracy. Soltani et al. [25] exploited physicochemical changes that occur during storage to investigate the possibility of non-destructive classification and quality inspection of eggs. Several machine learning techniques were developed for freshness recognition, including artificial neural networks, Bayesian networks, decision trees and support vector machines. All algorithms presented an accuracy of 100% with correlation coefficients up to 0.92. Nematinia et al. [26] designed a machine vision system based on several quality markers (yolk height, yolk and albumen density, HU, pH, egg weight). Visible images were processed to train an artificial neural network. The best results were obtained using HU and albumen pH as benchmarks, with correlation coefficients of 0.93 and 0.87, respectively.

Another young yet highly efficient and powerful technique for non-destructive testing (NDT) is active thermography [27]. In contrast to conventional passive thermography [28], it makes use of an artificial heat source to induce thermal contrast between defected and sound regions, and an IR camera with a high thermal sensitivity to measure the resulting surface temperature response in the stationary or transient regime [29]. In PT [30], the thermal energy is delivered to the object surface by means of one or more sudden heat pulses and its surface temperature evolution is monitored by acquiring thermograms in time sequence during the transient heating (cooling) phase. IR images obtained through PT are affected by local variation of the emissivity coefficient and by non-uniform heating of the surface which can significantly deteriorate the image quality, hiding sub-surface details. These drawbacks can be overcome by performing PPT [31]. The experimental setup for PPT is the same as in PT, and it basically consists of an IR camera, a flash heat source and a synchronization unit. The target is heated with a pulse, as in pulse thermography, and the mix of frequencies of the thermal waves is unscrambled by computing the Fourier transform of the temperature evolution over the field of view. The resulting phase, or magnitude, image can be presented as in modulated lock-in thermography. The fact that pulse phase thermography sorts available information coherently in term of frequencies brings interesting features with respect to the more traditional contrast approach used in pulse thermography [32,33].

Dealing with PT applications, Montanini et al. [34,35] designed a fast and automatic protocol to easily visualize the egg air chamber and evaluate its size by means of a dedicated image processing algorithm. Such elaboration was performed using a morphological operator with ‘white top hat’ transformation. Preliminary results, based on a limited number of eggs, showed a good correlation between infrared (IR) imaging of air chamber projection areas and egg weight loss (R^2^ = 0.96 and R^2^ = 0.92 for lateral and front projections, respectively) at 28 °C ± 0.1 °C and 30% ± 0.1% of RH.

The main aim of this work is to provide a new quality control tool that can be used to evaluate the freshness of eggs in a subjective and automated manner. Starting from the method introduced in [34,35], new features have been developed and introduced, with the purpose of improving the overall reliability. More in detail, in order to reduce flash reflections and improve the signal-to-noise ratio of the IR images, pulse phase thermography (PPT) was used additionally to PT. Thus, the discrete one-dimensional Fourier transform (DFT) was applied on each pixel of the thermogram sequence to compute amplitude and phase images. Phase images were used together with thermograms obtained from PT to enhance the accuracy of the sizing of the air chamber. The post-processing algorithm has been refined, too, allowing to compute the entire volume of the air chamber, other than to the projection areas in the lateral and in the front view. The ability of the proposed technique to measure the air chamber sizes was firstly assessed through specific repeatability and reproducibility tests. Finally, an experimental campaign was carried out on a homogenous and representative sampling made of 30 eggs, equally divided in three categories graded by weight (M, L, XL) and subject to an accelerated aging process. The possibility to directly measure the air chamber parameters and to monitor their evolution in time represent some important steps toward a better characterization of the egg aging process.

## 2. Materials and Methods

### 2.1. Experimental Setup and Geometric Calibration of the IR Camera

To carry out the experimental tests (Figure 1), an IR camera (mod. SC7600, FLIR Inc., Wilsonville, OR, USA), working in the mid-wave infrared range (MWIR 3.6–5.1 μm), was used. The IR camera has a spatial resolution of 640 × 512 pixels and a noise equivalent temperature difference (NETD) < 20 mK at room temperature. It was equipped with a 50 mm lens and fixed onto an optical table at a distance of about 40 cm from the egg. To heat the target, a xenon toroidal flash of 3 kJ was employed, which typically produced a very small overheating of the egg (less than 1 °C) for about 10 ms. The flash was interposed between the camera and the egg, as to ensure a sufficiently uniform heating of the region of interest. A control unit (Edevis GmbH, Stuttgart, Germany) was also needed to synchronize thermal pulse generation with IR image recording. A special support was realized to hold the egg and to allow it to be rotated precisely by 90° around its transversal axis in order to capture both the front and the lateral aspect of the egg.

A preliminary 2D camera calibration was carried out in order to convert image pixels into metrical units and remove the aberration of the optics [36]. For this purpose, a calibration grid was manufactured using an aluminum plate and machining a regular grid of circular slots. The circular slots had a 4 mm diameter and a center-to-centre pitch of 10 mm. The calibration grid was mounted in the same position of the egg and temperature and phase images were recorded (Figure 2). Image correction was applied using the proper function implemented in the National Instruments Vision Assistant software.

### 2.2. Testing Protocol

To reproduce an unbiased and realistic situation, a sample of 30 brown eggs was purchased by the same producer, without any specific inclusion or exclusion criteria. The producer was asked to provide 10 eggs for each of the three size categories indicated by the marketing standards [9]: extra-large eggs (XL, weight > 73 g), large eggs (L, weight between 63 and 73 g) and medium eggs (M, weight between 53 and 63 g). The egg weight was verified by the use of an analytical balance.

To reduce the uncertainty related to uncontrolled storing conditions, throughout the testing campaign, all the eggs were kept in a climatic chamber at a temperature of 28 °C ± 0.1 °C and with relative humidity RH of 30% ± 0.1%. These environmental conditions, which are quite more severe than those usually recommended for conservation (15–18 °C, RH 40–70%), were deliberately chosen to accelerate eggs aging [37]. At the same time, the use of controlled laboratory conditions does not represent an issue for the general validity of the method, since the quality control operations for eggs usually occur in indoor environments, where the effect of external influences is minimal.

The duration of the egg investigation was chosen on the basis of two criteria according to the European Commission regulation [9]:
the start took place 3 days after the laying, i.e., the day successive to the purchase. In fact, the eggs can be marketed within 4 days of their laying date.the end was established 31 days after the laying. The analysis was deliberately extended 3 days after the expiration date (28 days after the laying) in order to identify a trend boundary.

At regular intervals, the eggs were removed from the climatic chamber, stabilized at their surface temperature of 27 °C ± 0.2 °C, weighed with an analytical balance, placed on the sample holder and hence tested. Moreover, egg aging is a relatively slow phenomenon, therefore it does not require a high sampling. The whole set of images of the heating and the cooling phases was recorded with a frame rate of 100 Hz. For PT (Figure 3), only two thermograms, captured at time t = 600 ms and related to two orthogonal views (namely front and lateral), were used. Instead, for PPT, DFT was computed on all thermograms of the cooling phase, obtaining a magnitude and a phase image for each of the two orthogonal views. Actually, only the last one was used to present the relevant information about egg aging. Furthermore, it is important to underline that the IR images were acquired considering an emissivity coefficient *ε* = 1. Unlike traditional investigations [38,39], here we focused on the infrared signature produced by the different heating of the air contained in the air chamber with respect to that of the organic compounds of the egg and therefore, radiometric information were no longer needed [28].

Finally, PPT images were processed by a dedicated algorithm, developed in MATLAB (MathWorks, Portolla Valley, CA, USA). The latter is able to automatically compute both the area of the front and lateral views of the egg and the volume of the air chamber, by segmenting the images, based on a determined heuristic threshold.

### 2.3. Repeatability and Reproducibility Tests

Specific tests were also carried out in order to investigate the robustness of the proposed inspection method, applying the specifications reported in [40]. These tests were performed on a total of nine eggs (three for each of the three classification categories), at specific daily intervals from the 8^th^ to the 18^th^ day after laying. In particular, the following protocol was adopted:
according to the testing procedure reported in Section 2.2, five thermograms were acquired, ensuring that before each acquisition the surface temperature of the egg was always 27 °C ± 0.2 °C (repeatability test);according to the testing procedure reported in Section 2.2, a single thermogram was acquired and the egg was repositioned in the climatic chamber. After one, two, three and four hours, the whole process was repeated (reproducibility test).

## 3. Results

### 3.1. Raw IR Images

Figure 4 reports a selection of raw IR images recorded after PT (egg XL#7). The images are presented in gray tones, based on digital levels. The shape of the air chamber is pseudo-circular in the front view and like an ovoid cap in the lateral one. Thus, the air chamber volume could be geometrically approximated by the sum of two volumes, a spheroid and an ellipsoid.

Some critical issues linked to PT are clearly visible in the images. First, there are consistent reflections due to the flash, which affect thermograms taken both from the lateral view and from the front view, the latter being more problematic as they partially overlap the air chamber. Second, while the edge of the external contour of the egg is properly defined, that of the air chamber is not, as it shows some blurring along the outer edge of the egg. This effect is more pronounced in the lateral view.

Another aspect that is worthy of consideration is related to the background temperature. Due to the long duration of the tests (28 days) and the consequent variation of the ambient temperature, it was different from one test to the other (i.e., the black pixels of the background do not correspond to the same temperature). To improve the contrast between the background and the egg, we chose to adapt the scale of digital levels of each image instead of using a fixed amplitude scale.

### 3.2. Phase IR Images

Figure 5 shows infrared phase images obtained after PPT, for the same egg (XL#7) and the same views (front and lateral) displayed in Figure 4. In this case, each gray level refers to a specific value of the phase (in degrees).

It is noteworthy to highlight that the phase images allow the air chamber to be defined better, showing a well-contrasted edge. On the other hand, the egg shell is blurred and this prevents a clear definition of its outlines. In a practical sense, non-uniform heating and surface emissivity variations have a negligible impact on phase [41]. For this reason, reflections in the IR spectral band, which are clearly visible in Figure 4, are substantially reduced and appear in the phase image as black spots of limited sizes.

As the external edge of the egg is better defined in the raw image, while air chamber edges are sharper in the phase image, we adopt a mixed approach here, which uses both sets of images.

### 3.3. Image Processing for Air Chamber Sizing

Figure 6 reports the block diagram of the algorithm used to measure the relevant geometrical parameters of the air chamber, as the front view is considered.

Initially, the software segments both the raw and the phase images: for the first one, it separates the egg from the background, while for the second one, the air chamber projection from the egg; in both cases, it uses a binarization process. After that, a geometric analysis of the air chamber projection takes place. The rectangular region, which circumscribes the segmented egg in the raw IR image, is applied to the phase image. This creates a new, hybrid and resized image with a dark and homogeneous section around the air chamber projection. Hence, the two-dimensional shape of the air chamber projection can be well approximated by an ellipse.

Figure 7 illustrates the procedure, performed by the presented algorithm, to identify the points on the perimeter of the air chamber projection in the aforementioned, hybrid and resized image.

All pixel rows that cross the air chamber projection are fitted with a smoothing spline in order to improve the accuracy of the perimeter measurement. Considering the portion which goes from the first pixel of the row to that associated to the x-coordinate of the center of the air chamber projection (blue dotted line), the point *P1* is identified as the last pixel having a brightness level lower than a threshold *w_th_*, equal to one third of the maximum brightness of the row profile *b_max_*. The threshold value is determined heuristically and fits well with all the analyzed cases. In the same way, the point *P2* is defined on the portion which goes from the pixel associated to the x-coordinate of the center of the air chamber projection to the last pixel of the row (red dotted line).

Since the minimum number of points to fit a shape in an ellipse is five, it is necessary to repeat the procedure for multiple rows or columns. Typically, a high number of distributed points allows to obtain a more robust fitting. Once the parameters of the equation of the ellipse are known, it is possible to calculate the two semi-axes of the ellipse, *a* and *b*, which represent the characteristic sizes of the air chamber projection from the front view. An example of air chamber size measurements, using the above explained technique, is shown in Figure 8a. Furthermore, a similar approach is adopted for the lateral view of the egg in order to estimate the height of the air chamber, *c*. The same steps, computed for the front view, are repeated but in this case, the algorithm was developed to identify the two profiles that surround the air chamber projection along its depth. The more external profile (red line in Figure 8b), which corresponds with the egg perimeter, is detected by considering for each row the last point greater than zero (in the first part of the processing, the brightness of the background pixels was set to zero). Instead, the internal profile (green line in Figure 8b) is focused by considering for each row the last value lower than a certain threshold. Since the air chamber occupies a well-defined region (dull pole), the area for the analysis is limited to the group of rows having the maximum x-coordinate in the external profile. Air chamber height for each image row is estimated as the x-coordinate difference between the external and the internal profile. The parameter *c* is then computed as the average air chamber height in the region around the dull pole.

According to the previous consideration, the air chamber area *A_b_* is expressed as that of the fitted ellipse *A_b_ = πab*, with the original value in pixel converted in mm^2^ by a scaling factor obtained from the camera calibration process. Unlike the computation of the area, that of the air chamber volume *V_b_* is affected by two main problems. The first one is that the volume data present a significant uncertainty due to the complications in the estimation of the air chamber height I. Secondly, two image acquisitions per egg (i.e., front and lateral view) are required for the evaluation of the three air chamber parameters (*a*, *b* and *c*). For these reasons, the air chamber volume was approximated by means of the parameter *A_b_*^3/2^. Such quantity has the same dimensions of a volume and it was estimated starting from the measurement of the projection of the air chamber area, using only a single image acquisition (i.e., egg front view) per egg.

### 3.4. Measurement Repeatability and Reproducibility

Figure 9 reports the results of the repeatability (first column) and reproducibility (second column) tests, as images taken from the front view are considered. The daily mean value of the area of the air chamber and its relative standard deviation, considering a 68% coverage interval, were computed for each sequence of images for each egg.

All plots show a monotonic increase of the air chamber area with the elapsed time, as a result of the enzymatic decomposition process of the organic components of the egg. However, it is important to highlight that the growth of the air chamber occurs with different rates, even for the case of eggs belonging to the same category. As regards to the uncertainty related to the repeatability and reproducibility tests, it can be observed that the standard deviation never exceeds 5% of the measurement.

Table 1 shows the value of the daily standard deviation expressed as a percentage of the daily mean of the area of the air chamber.

The missing data in Figure 9 and Table 1 are associated with errors in the air chamber sizing process. This mainly occurred when PPT produces reflections on the egg shell and when the air chamber is close to the egg perimeter (Figure 10a). In such a situation, there is no longer a sharp transition between the air chamber and the egg in the row brightness profile, making the identification of air chamber perimetral points potentially inaccurate. In other cases, errors in the estimation are related to the physical structure of the air chamber itself, e.g., holes in the air chamber caused by shell defects (Figure 10b).

Furthermore, a large part of the tests demonstrated as the front image allows to achieve a more reliable measurement of the air chamber size. This result is not strictly related to the algorithm performance, but rather to the air chamber shape and position and to the quality of raw images. As it can be seen from Figure 11, the air chamber profile is sometimes not visible in the lateral view or, in other cases, it appears very irregular, making the estimation of its height virtually impossible. For this reason, no statistical results were reported for the lateral view.

### 3.5. Estimation of Eggs Freshness

To provide a general trend of the evolution of the air chamber size as time passes, the geometrical parameters of the air chamber were derived from the statistical analysis of the corresponding measurements performed on 10 eggs of the same category.

In Figure 12, the mean values ($\bar{a}$, $\bar{b}$ and $\bar{c}$) of the geometrical parameters *a*, *b* and *c* are plotted, for each egg size category, as a function of the elapsed time. The uncertainty bands are expressed in terms of standard deviations.

$\bar{a}$ and $\bar{b}$ exhibit a similar behavior, with a first part that is almost linear and a second part, which starts after the 18th elapsed day, that flattens out with increased dispersion of the experimental data. On the other hand, $\bar{c}$ displayed a more linear trend with uniform dispersion.

Figure 13a shows the reduction of the egg weight with time. The curves start at the first day of testing (i.e., 3 days after laying), which is used to normalize the ordinate scale. For all the three egg categories, the trend is quite linear with an increased dispersion of the experimental data with time. The slope of the curves is dependent on the egg category: bigger eggs are affected on average by the fastest weight loss, but they also displayed the highest variability. Instead, in Figure 13b, normalized mean values of the air chamber area (i.e., *A_b_*/*A_b_*_0_) are compared. The increment of the air chamber size is evident in all the plotted curves, even though it does not seem to exist a clear correspondence between increasing rate and egg size. Furthermore, bigger eggs exhibit a more irregular trend and higher data dispersion, especially for longer elapsed times.

These results demonstrate that a specific size of the air chamber for the single category of eggs is quite difficult to be defined. In fact, as shown in Figure 12, the air chamber sizes increase with the elapsed days, but their trends remain substantially similar between the different categories. Furthermore, as reported in Figure 13, the standard deviation of the air chamber sizes significantly increases even for eggs of the same category. Further investigations or the identification of the initial air chamber size could be desirable.

The previous results suggest a correlation between egg weight reduction and air chamber size increase. To verify this, in Figure 14, mean values of the normalized air chamber area are plotted as a function of the mean value of egg weight loss. A second order polynomial regression function (*y = β*_0_
*+ β*_1_*x + β*_2_*x*^2^) describes the quantitative relationship between these two variables. The eggs show a different trend depending on the dimensional category they belong to, but with correlation coefficients R^2^ that were always high. Moreover, in this case, it can be observed that XL eggs are characterized by a more pronounced variability.

Finally, Figure 15 shows the relationship between the normalized air chamber volume (i.e., *V_b_*/*V_b_*_0_) and egg weight loss.

The dispersion of the experimental data is sufficiently small as the M and L size categories are considered, while, again, it is appreciably greater for the XL one. In such case, it would seem that the approximated expression *A_b_*^3/2^ is no longer able to return the correct value of the air chamber volume after the 16th day of testing. This could be due to the excessive growth of the air chamber and its approach to the egg perimeter, which do not allow the correct computation of *a* and *b* in the front view.

## 4. Discussion

The results reported in the previous sections are based on the use of IR images to measure the air chamber size of an egg, which has proven to be a relevant parameter to assess the egg freshness. In order to do that, a hybrid approach that makes use of both the phase image obtained by PPT and one thermogram recorded during the cooling stage has been developed. This method allowed to measure the main geometrical parameters of the air chamber with better accuracy, thanks to the greater contrast and the consequent better definition of the edges. Moreover, it should be noted that some sizing errors may still occur due to the proximity of the air chamber to the egg perimeter. In these situations, it is not possible to determine the air chamber borders correctly and approximate the area with a regular geometry. The measurement of both the front area of the air chamber and the egg weight over time revealed the presence of interesting findings:
the existing trends are category-dependent;data scattering increases with the elapsed days for all the egg categories;bigger eggs seem to display more irregular behaviors, as the variability of the shape and position of the air chamber is more pronounced as the egg size increases;from Figure 12, it is possible to observe how, for all the egg categories, the air chamber height (*c*) never exceeds 4 mm up to 9 days after laying. This is in agreement with the definition of “extra-fresh” eggs provided in [9];the plot of the egg weight reduction versus the increase of the air chamber area indicated the presence of a linear relationship with a high degree of correlation (R^2^ > 0.98).

To study the influence of all the air chamber sizes, the air chamber volume has also been investigated. Here, two specific image processing problems were identified. The first was that volume data are characterized by higher uncertainty due to the difficulties in estimating the air chamber height (*c*). The second was that, for the evaluation of the three parameters of the air chamber (*a*, *b* and *c*), two acquisitions (i.e., front and lateral views) were necessary. Such aspect made the whole procedure more complicated, without introducing significant improvements in terms of uncertainty reduction of the obtained results. As a consequence, it is possible to conclude that the monitoring of the front area of the air chamber provided a sufficient adequate, inexpensive and fast way to evaluate the egg aging process.

## 5. Conclusions

The present work deals with the development of a robust and automated approach to quantitatively determine egg freshness. It is based on a hybrid method, which combines both PT and PPT to grab the air chamber images without causing any aesthetic, structural or organoleptic damage to the egg itself. The acquired raw and phase IR images have been post-processed using a segmentation algorithm with heuristically defined threshold values, which allows to calculate all the relevant geometric parameters of the air chamber. Repeatability and reproducibility experiments have confirmed that the proposed method is a reliable tool to estimate the air chamber size, being the measurement uncertainty always lower than 5%. Therefore, the technique has been further tested on a larger sample of 30 eggs, equally divided in three categories graded by weight (M, L, XL) and subject to aging.

Results have shown a monotonous increase of both the air chamber projection areas and the air chamber volume with egg aging. This increase of the air chamber is related to the enzymatic decomposition process of the organic components of eggs and, therefore, to the weight loss, which represents a well-known indicator of egg freshness. In agreement with that, an excellent correlation between the normalized values of the air chamber area seen from the dull pole and the egg weight reduction has been found. Eggs belonging to different size categories behave differently: the correlation coefficients vary from a minimum of 0.986 for XL eggs to a maximum of 0.998 for M eggs. As far as the air chamber volume is considered, the correlation found was less strong, with R^2^ ranging from 0.955 for XL eggs to 0.993 for M eggs. In any case, this does not represent a severe limitation, since it is certainly more practical and faster to restrict the observation to the front view only.

The proposed method, based on IR vision, might be very promising for application in the poultry industry and in sales chains, as it provides an effective and quick tool for monitoring non-destructively large batches of eggs. For example, eggs could be analyzed after storage at controlled temperature and humidity. Their handling could be ensured by a robotic arm, with the ability of simultaneously grasping several eggs and of guaranteeing an optical access to their two views, front and side. After the thermographic investigation, the IR images should be processed by using the described algorithm for air chamber sizing. Additional software and equipment could integrate the various procedures and highlight/exclude the eggs that do not respect the parameters of suitable freshness. Furthermore, for this application, no specific lighting conditions are required and, depending on the available resolution of the IR camera, the whole process can be easily extended to run simultaneously on a large number of eggs. However, in order to implement the presented methodology in the poultry context, users should take into account several factors [42,43,44]. First of all, the ambient parameters, i.e., relative to the place where the assessment is carried out (e.g., humidity and temperature), should be analyzed. Subsequently, the technical issues, i.e., relative to the equipment to be used, should be assessed. Therefore, for a correct evaluation procedure, the proposed method should be applied in a specifically designed and controlled environment. In addition, the emissivity of the egg surface can be considered sufficiently constant close to the ambient temperature. It does not change with air chamber size, because it depends on the temperature and on the geometry of the shell. In any case, the use of PT and PPT ensures that the absolute temperature is low relevant, but rather the variations in temperature (i.e., the contrasts of gray tones in the IR images) are of greater interest. Lastly, since the air chamber is a common parameter to all eggs, the proposed method could be applied to assess the freshness of any type of egg. However, it must be specified that freshness depends on the different biology of these foods (size, organic components, etc.).

In conclusion, PT and PPT are NDTs and, for this reason, they can be employed on the whole production without any loss. Furthermore, unlike other NDTs, they ensure that the biological and organoleptic characteristics of the eggs remain unaltered. Finally, they allow to obtain a good repeatability and reproducibility, a low measurement uncertainty, as well as guarantee an automatic and objective investigation process. The only disadvantage that can be counted is the initial cost of the suitable equipment, which includes a high resolution infrared camera. In any case, this is easily compensated by the certainty of obtaining a high quality of the eggs.

## Figures and Tables

**Figure 1 sensors-22-03496-f001:**
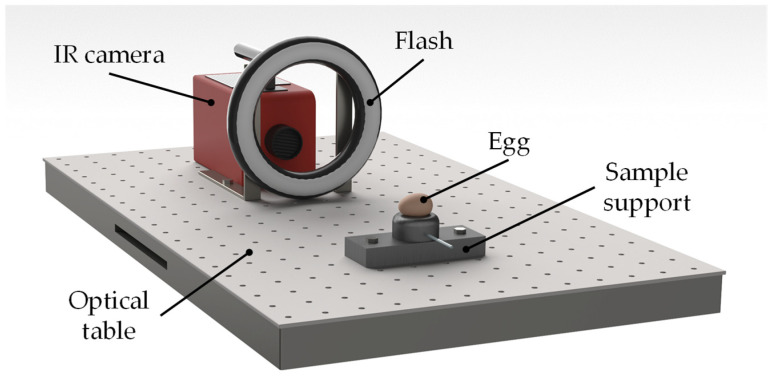
Schematic representation of the measuring principle of pulse phase thermography.

**Figure 2 sensors-22-03496-f002:**
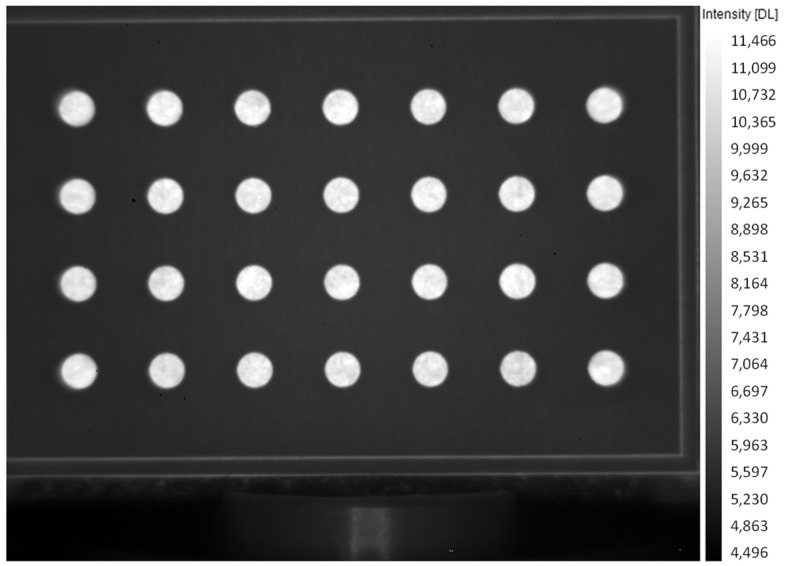
IR image of the calibration grid (units are in digital levels, DL).

**Figure 3 sensors-22-03496-f003:**
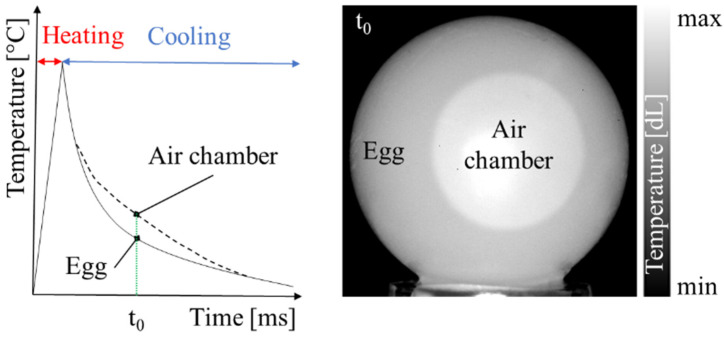
Conceptualization of PT in the presented method.

**Figure 4 sensors-22-03496-f004:**
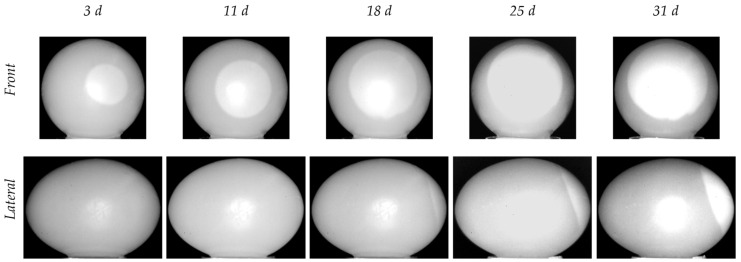
Grayscale raw IR images of egg XL#7 recorded on different days. First row: front view (dull pole). Second row: lateral view.

**Figure 5 sensors-22-03496-f005:**
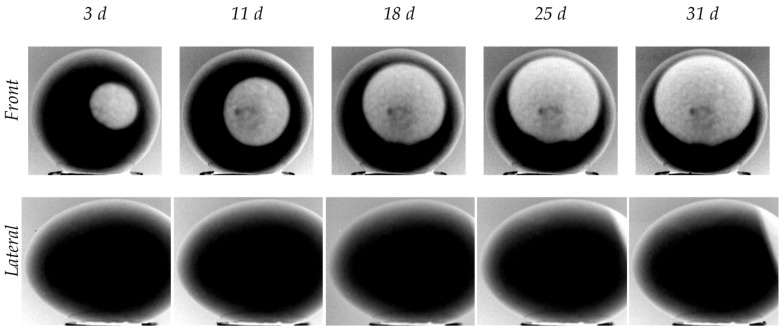
Grayscale phase IR images of egg XL#7 recorded on different days. First row: front view (dull pole). Second row: lateral view.

**Figure 6 sensors-22-03496-f006:**
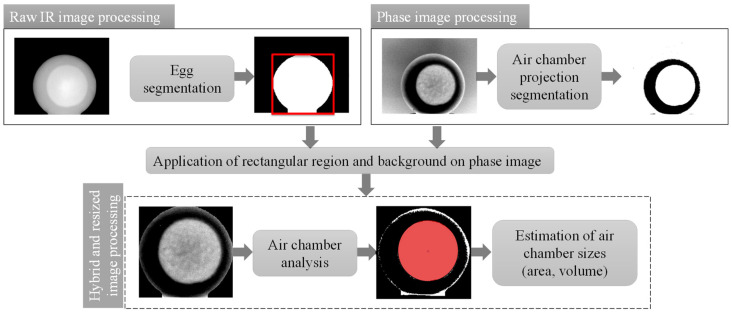
Block diagram of the algorithm used to estimate air chamber size from the front view using both raw and phase infrared images.

**Figure 7 sensors-22-03496-f007:**
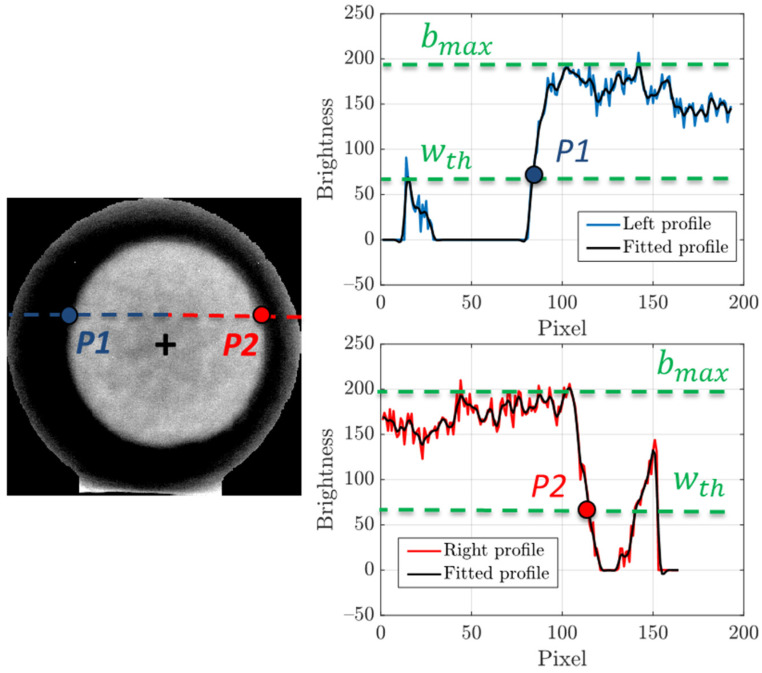
Identification of key points of the fitting ellipse for the air chamber perimeter.

**Figure 8 sensors-22-03496-f008:**
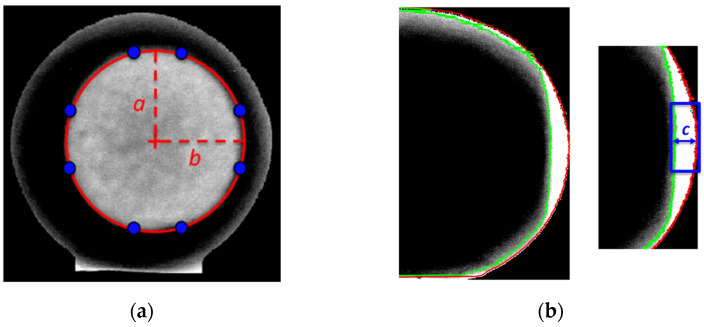
(**a**) Fitted ellipse and estimated semi-axis sizes (*a*, *b*) using eight data points (front view) and (**b**) external and internal profiles used for estimation of the air chamber height (*c*).

**Figure 9 sensors-22-03496-f009:**
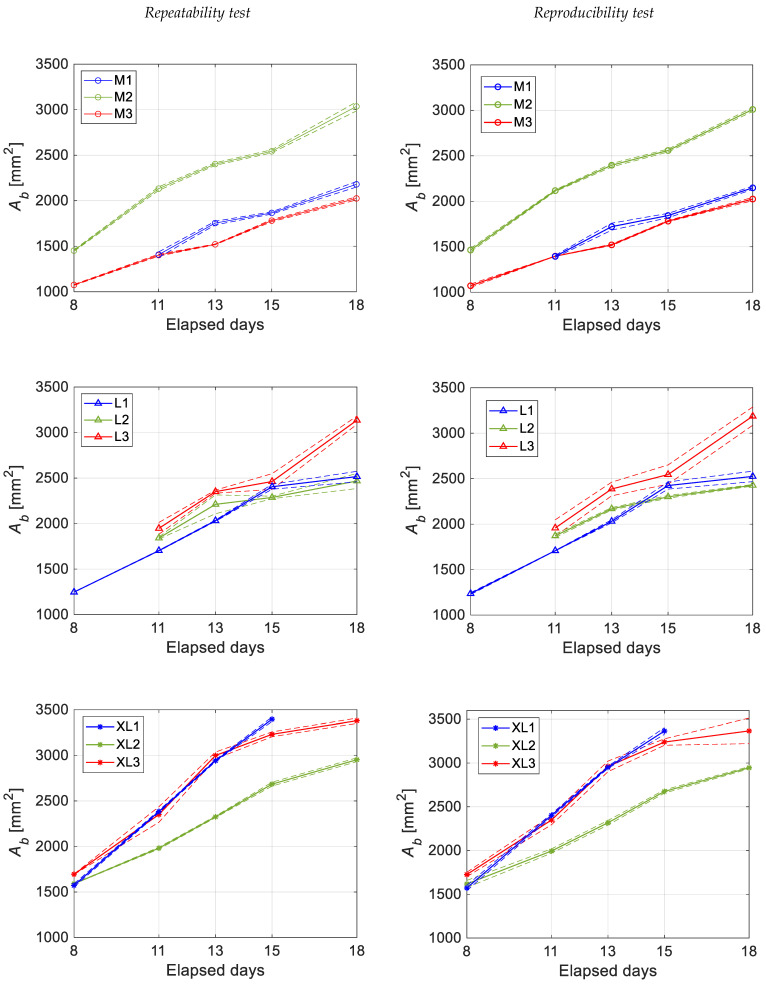
Daily mean value, estimated with a 68% confidence interval, for repeatability and reproducibility tests of the area of the air chamber for different egg categories (M, L, XL).

**Figure 10 sensors-22-03496-f010:**
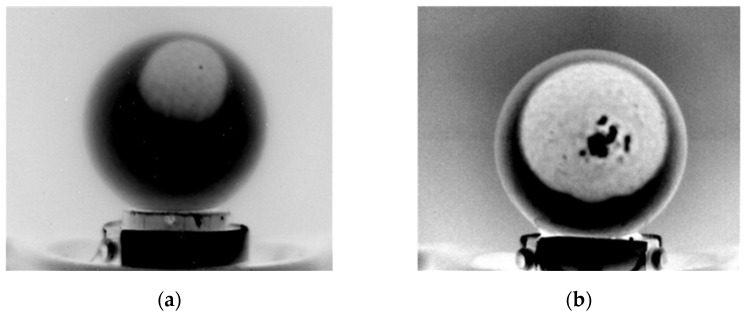
Causes of errors in air chamber sizing: (**a**) reflections on eggshell and (**b**) holes in the air chamber (defects of the egg shell).

**Figure 11 sensors-22-03496-f011:**
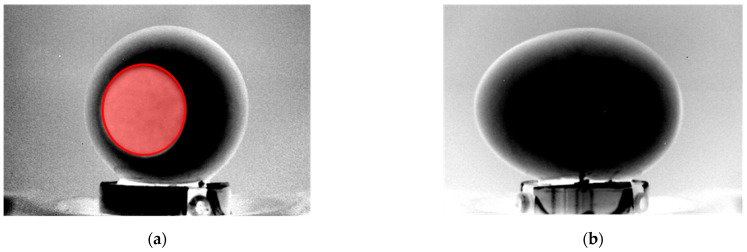
(**a**) Front and (**b**) lateral views of egg M#9: in the lateral view, the air chamber does not appear.

**Figure 12 sensors-22-03496-f012:**
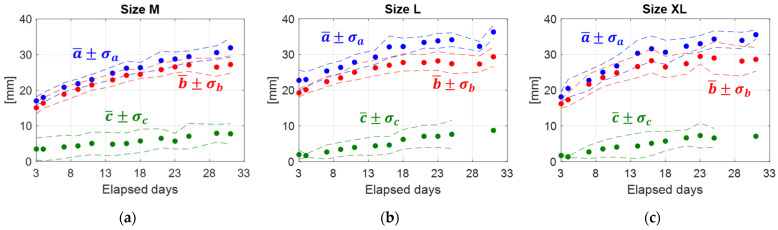
Estimation of the mean values (filled dots) of the geometrical parameters of the air chamber at elapsed days for egg of size M (**a**), size L (**b**) and size XL (**c**). $\bar{a}$ and $\bar{b}$ are the mean values of the minor and the major semi-axis of the air chamber on the front view, respectively, while $\bar{c}$ is the mean value of air chamber height on lateral view for each egg size category. σ_a_, σ_b_ and σ_c_ are the standard deviations of the previous defined quantities.

**Figure 13 sensors-22-03496-f013:**
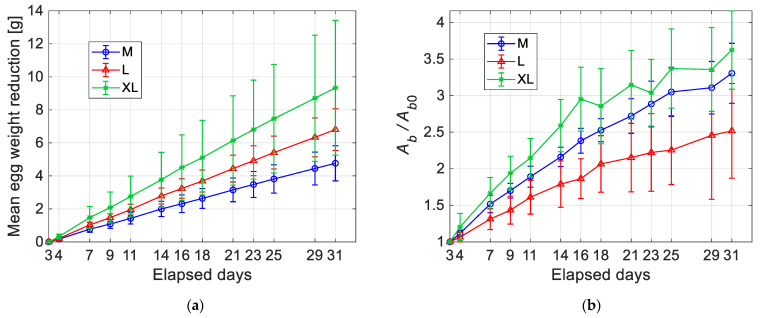
Mean values of the egg weight reduction (**a**) and increment of the mean values of the normalized air chamber area (**b**) vs. elapsed days for different egg categories (M, L, XL).

**Figure 14 sensors-22-03496-f014:**
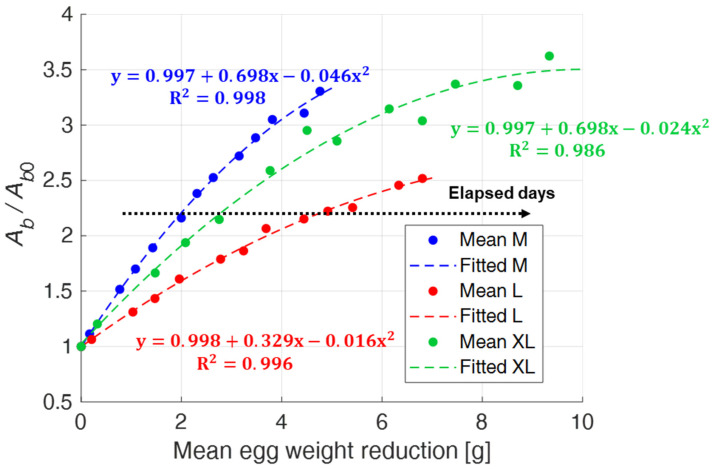
Second order polynomial regression between mean values of the normalized air chamber area and mean values of the egg weight reduction for different egg size categories (M, L, XL).

**Figure 15 sensors-22-03496-f015:**
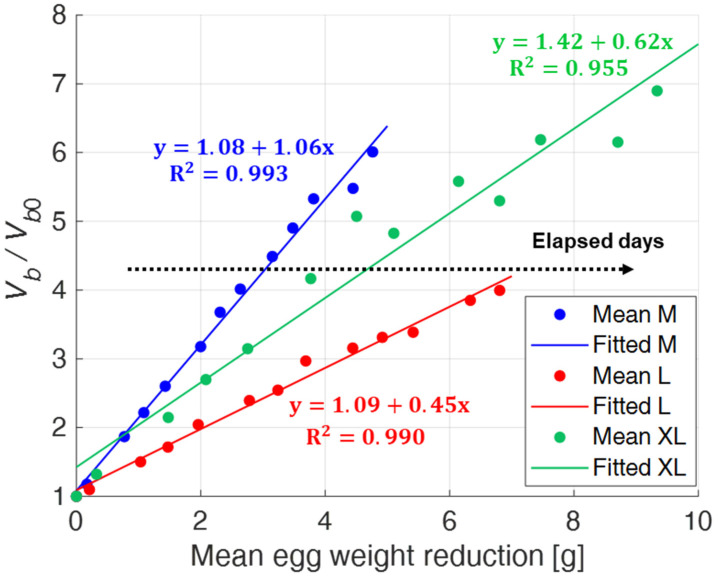
First order polynomial regression between mean values of the normalized air chamber volume and egg weight reduction for different egg size categories (M, L, XL).

**Table 1 sensors-22-03496-t001:** Standard deviation of the measured area of the air chamber for repeatability and reproducibility tests.

Repeatability Test—St. Dev. (%)						Reproducibility Test—St. Dev. (%)					
# Egg	Elapsed Days					# Egg	Elapsed Days				
	8	11	13	15	18		8	11	13	15	18
M1	N/A	2.14	1.14	0.67	1.36	M1	N/A	0.87	2.22	1.31	0.80
M2	0.58	0.92	0.59	0.65	1.63	M2	1.29	0.45	0.79	0.64	0.70
M3	0.47	0.78	0.24	0.65	0.82	M3	1.70	0.09	0.51	0.40	0.88
L1	0.21	0.24	0.34	1.13	2.29	L1	0.80	0.14	0.81	1.62	2.29
L2	N/A	0.62	4.68	0.60	3.35	L2	N/A	0.96	0.62	0.65	0.35
L3	N/A	3.32	0.72	3.51	1.44	L3	N/A	4.57	3.15	4.15	3.15
XL1	1.12	0.52	0.35	0.72	N/A	XL1	1.91	0.74	0.43	1.13	N/A
XL2	0.13	0.52	0.35	0.79	0.70	XL2	2.50	1.23	1.13	0.69	0.43
XL3	0.15	3.63	1.25	0.82	0.85	XL3	1.46	2.29	2.05	1.13	4.32

## Data Availability

Not applicable.

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
