# Peer review of "Development, Validation and Preliminary Experiments of a Measuring Technique for Eggs Aging Estimation Based on Pulse Phase Thermography"

_sensors, 2022, doi:10.3390/s22093496_

Round 1

Reviewer 1 Report

The article deals with an interesting topic. It seems to be a simple problem, which, after careful consideration, shows that it is relatively complex. The authors used a system of non-destructive examination, application of electromagnetic radiation in the infrared region, for qualitative and quantitative evaluation of the area of interest.

The article is written logically, the information is clear and understandable to the reader. I have no comments.

Author Response

Dear Reviewer 1,

thank you very much for the time and the effort that you spent to provide your valuable feedback and for appreciating our work.

Reviewer 2 Report

The manuscript is very interesting, I think the idea is very good.
The problem for me is that the measurements were done with 10-10 different sized eggs. I find this sample number very low for such a complex sample. Also the size range of M, L and XL size eggs is very wide, I would definitely recommend to increase the sample size. If this is not possible, at least include the term "preliminary experiments" in the title.
-The size of the air chamber was used to estimate the spoilage/freshening status - but no statistical comparison of the data was done. 
-What was the size of the air chambers of the samples judged to be the same size? Identical? Did it differ? 
-The table shows 3-3 samples - where did the other measurements go?
-How many duplicate measurements were made?
- It is concluded that the method is suitable for automatic measurement. Please explain this in one or two sentences - how do you envisage this to be achieved:

Formatting problems:
- The title of Table 1 is incorrect.
- There are two chapters 4

Reviewer 3 Report

Infrared thermography is a technique that has been widely used in areas such as engineering and medicine to assess changes in the amount of heat radiated from an object. The application of IR to food processes such as egg production represents a tool that could improve the quality of eggs destined for human consumption. This article shows an innovative application of thermography to assess egg quality, which is of great help to companies.

The lack of a clear objective and a discussion that supports the results obtained are some deficiencies that I noticed in the article. I have left some comments and suggestions that may help the authors to enrich their manuscript.

Lines 10-25: In the abstract, clearly state the objective of the study.

Line 41-42: The 6 mm that the authors describe applying for eggs for all species? For example, hens, ducks, quails, or other birds? I consider it necessary to specify this since the size and even organic components can differ.

Response:

Line 43 Please, include a brief description of the term “extra fresh” egg.

Line 44: Please indicate the measure to consider an air chamber as “small”.

Line 49: I would suggest including a small paragraph regarding the importance or the main health/economic issues associated with the freshness of eggs. Why is it relevant to evaluate the freshness of an egg?

Line 107: Only the visualization of the egg air chamber is used as a method to evaluate the quality of the egg? Or there are some other indicators? Please, discuss this.

Line 112: In the mentioned results, the authors have found that environmental factors such as humidity or environmental temperature alter the thermography values? If so, I think it is necessary to include this information.

Line 113-124: Please, clearly state the aim of the study in the introduction. According to these lines, it seems that the study has three objectives, but they are not properly described. Additionally, there is no hypothesis for the study.

Lines 128-146: I suggest moving this paragraph to the Introduction section, since Materials and Methods describe how the experiment was performed.  

Line 172: Please indicate the method to calculate the sample size. Also, please include if the eggs came from the same producer or if authors had inclusion or exclusion criteria for the collected eggs.

Line 180: Include the criteria to determine the measurements days of each egg.

Lines 191-192: For a better understanding of this part of the methodology, it may be appropriate to include a picture to illustrate how the measurements were performed, or if the authors used a determinate color palette.

Line 319-322: It is suggested to delve into the process by which the enzymatic failure of organic components occurs and why it is different even in eggs of the same category.

Line 402: At the beginning of the discussion, it is recommended to include a brief introduction highlighting the relevant findings of the authors. For example, the association between the weight, the size of the air chamber and the freshness of the egg. Since the main objective of this study was to use IR to evaluate these elements, it would be appropriate to highlight this.

Line 404: Since the applicability of pulsed phase thermography to determine egg freshness is a novel issue and there may not be enough available data to compare and discuss, the authors could analyze the factors that can influence the IR readings. For example, environmental parameters such as humidity, environmental temperature, or technical aspects of the camera could alter the results. Some articles that may help the authors are listed below:

Doi: 10.1016/j.infrared.2015.02.007

Doi: 10.3390/ani11082247

Doi: 10.3390/ani12060789

Line 407: According to the obtained results, the authors could give a range of normal parameters that researchers could use as basis un further investigations?

Line 409: The air chamber size changes could modify the degree of radiation from the object? Please, discuss this.

Line 413-421: According to your observations and suggestions, could you explain why monitoring the size of the air chamber and the analysis of the front view of the egg would be sufficient to evaluate freshness?

Line 422: The authors mentioned that they used more severe environmental conditions for storing the eggs. Does the author think this could influence the results? If this could be a limitation of the replicability of the results, I would consider it important to mention it.

Line 423: I would suggest rephrasing the conclusions according to the objective that must be described in the abstract and the introduction.

Reviewer 4 Report

Assessment of the freshness of eggs destinated to human consumption is an extremely important goal for modern food industry and sale chains, as eggs show a rapid natural aging which also depends on the storage conditions. Traditional techniques, candling and visual observation, have a low level of automation and a subjective and qualitative nature of the analysis. In this paper, pulse phase thermography has been exploited as a fast (few seconds) and effective method for monitoring the size of internal air chamber of the eggs and for estimating the freshness of this food. Raw infrared images, i.e. original data grabbed by the infrared camera, have been post-processed in the frequency domain by computing the discrete Fourier transform and the resulting phase images have been then used to infer eggs aging over time. The manuscript begins with a rich passage through current research, the authors have shown that there is ample documentation on this research. It continues with a comprehensive description of the techniques used in both hardware and software. The analysis is well defined and described. The authors should identify which technique they have designed is better, what are the advantages and what would be the costs of implementing such a system in the commercial market. The reliability of the method in assessing egg freshness is satisfactory, as repeatability and reproducibility tests proved that the uncertainty of air chamber size estimation never exceeds 5%.

Round 2

Reviewer 2 Report

I accept the revised and corrected manuscript for publication

Reviewer 3 Report

The authors have considered each of my suggestions despite being too many.
The article is now clearer and more comprehensive. They have done a great job.
That's why I suggest it should be published.

Respectfully